# Young mothers' attitudes towards domestic violence and their maternal healthcare services utilization in Bangladesh: A multilevel cluster analysis

Sawkia Afroz[1], Tasmiah Sad Sutopa[2], Md Rabiul Haque[1]*

1 Department of Population Sciences, University of Dhaka, Dhaka, Bangladesh, 2 Department of Statistics, University of Dhaka, Dhaka, Bangladesh

* rabiuldps@du.ac.bd

## Abstract

This paper examined the association between young mothers' attitudes towards domestic violence and four or more antenatal care (ANC) and health-center-based delivery service utilization using two cross-sectional waves of the Bangladesh Demographic and Health Surveys (2014 and 2018) data. We carried out a multilevel logistic regression analysis. Findings show that a strong cluster variation exists in four or more ANC and health-center-based delivery service utilization. Although the utilization of four or more ANC and health-center-based delivery services has increased over the years, it is far behind the targets of SDGs, particularly for young mothers with justified attitudes towards domestic violence. Extension of maternity allowance coverage and motivational programs are important policy recommendations.

## Introduction

Assurance of healthy lives for all is one of the commitments of the Sustainable Development Goals (SDGs), the blueprint for building a better world for the next generation. In achieving equity in the health systems, a reduction in maternal mortality is greatly desirable which can be ensured by improving mothers' access to quality antenatal care (ANC) and health-center-based delivery care. The utilization of four or more ANC and health-center-based delivery care is an important WHO-recommended strategy to reduce preventable maternal mortalities and their morbidities [1–3]. Moreover, young mothers' have a greater risk of experiencing maternal mortality due to their limited access to four or more ANC and health-center-based delivery care compared to adult mothers [4–7]. Previous studies have identified several barriers of four or more ANC and health-center-based delivery service utilization related to demographic and socio-economic aspects of the population. Moreover, the association with behavioral factors, especially the attitudes of young mothers towards domestic violence and access to four or more ANC and health-center-based delivery care need to be explored as it has become a burning issue over the world nowadays. The international community is also

**Data Availability Statement:** Data files for this survey are open to access and use through the DHS program website. USAID. The Demographic and Health Survey ICF, Rockville,Washington, D.C.,

USA: United States Agency for International Development; 2020 [cited 2020 8 august]. Available from: https://dhsprogram.com/data/available-datasets.cfm?fbclid=IwAR1H7zsRnId8cmR2kwd8fD-p2XDP7gQrXAd-oOOzKQdHiq7VpLUMDPLxM2c.

**Funding:** The authors received no specific funding for this work.

**Competing interests:** The authors have declared that no competing interests exist.

committed to ending all forms of violence against all women and girls by 2030 under SDG-5 (Target 5.2) [8].

In 2019, countries participating in the International Conference on Population and Development (ICPD+25) initiated an integrated approach to attain triple zero in terms of unmet need for contraception, preventable maternal deaths, and gender-based violence and harmful practices [9]. Globally, 35.0% of women have endured sexual or physical violence by their intimate partners [10]. Around 23.2% of women from wealthy countries and 37.7% of women from South East Asian countries including Bangladesh have experienced various forms of violence at different stages of their life cycle [11]. Violence against women adversely affects their physical and mental health because abused women are mostly less capable to care for themselves and their offspring [12, 13]. Evidence also suggests that in developing countries such as Bangladesh miscarriage, preterm birth, stillbirth, low birth weight in newborns and maternal deaths followed by maternal morbidity are indirect consequences of domestic violence against women during pregnancy [4, 14–23]. Domestic violence against women causes disempowerment to them, particularly for young women [24, 25] which ultimately limits their access to four or more ANC and health-center-based delivery care.

The Government of Bangladesh has been implementing several initiatives such as a community-based skilled birth attendant program, maternity allowance (MA) program, basic and comprehensive emergency obstetric care, active management of pregnancy complications, etc. The goal of these initiatives is to improve maternal healthcare services utilization, targeting to reach four or more ANC for 50% of mothers and health-center-based delivery for 65% of them by 2022 [26] and four or more ANC and health-center-based delivery for 98% of mothers by 2030 [27]. Under the MA program, each poor woman received an amount of TK 800.0 equivalent to 9.52 US$ per month for three years since gestation to purchase health care and nutrient foods for themselves and also for their children [28]. Although the utilization rate has increased in 2017–18 (referred to 2018 onward) compared to 2014, only 47% of mothers in Bangladesh received four or more ANC and 49% of them delivered their last child's birth at a health center in 2018 [29]. Such a pattern of using four or more ANC and health-center-based delivery care in Bangladesh is largely associated with women's lower social position, lower education, poor financial status, and inadequate understanding of their rights [30–33]. While the unavailability, inaccessibility, and unaffordability are strongly correlated with lower utilization of four or more ANC and health-center-based delivery services among Bangladeshi women, the association of domestic violence with antenatal and delivery care services utilization is less studied in the context of Bangladesh [31, 34–36].

Socio-cultural norms and gender roles typically shape Bangladeshi women's attitudes towards accepting domestic violence [36, 37]. In Bangladesh, at least three in every five women experienced physical or sexual violence and one in five women justify wife-beating by their husbands [32, 38–40]. Therefore, understanding the effects of domestic violence on young mothers' four or more ANC and health-center-based delivery service utilization is important for developing effective maternal healthcare policies and interventions.

The association between domestic violence and the lower utilization of four or more ANC and health-center-based delivery care receives less attention in previous research. Earlier studies have focused on either utilization of any ANC or violence caused by an intimate partner [36]. Moreover, the situation of ANC and health-center-based delivery care for young mothers, particularly those aged 15–24 years, remains unexplored as previous studies mostly considered women of all age groups [25, 36–38]. Also, the variations in the use of four or more ANC and health-center-based delivery by clusters are not considered in the past literature. In the context of Bangladesh, to the best of our knowledge, no study has considered domestic violence toward young mothers and its implication for using four or more ANC and health-

center-based delivery care nor appropriate statistical models have been used to reveal the cluster variations [41]. Moreover, the uniqueness of this study lies in using the recent two waves of BDHS data (2014 and 2018) for comparing the utilization of four or more ANC and health-center-based delivery services by women's attitudes towards domestic violence in Bangladesh. The findings of this study may contribute to higher ANC and health-center-based delivery services utilization in developing countries like Bangladesh, which contains a large portion (28%) of ever-married young women [29]. More specifically, a better understanding of the effects of relevant covariates along with cluster variation which is examined in this study may assist the policymakers in better executing the intervention programs targeting to increase the utilization of WHO-recommended four or more ANC and health-center-based delivery services for young mothers.

## Materials and methods

### Data source

The Demographic Health Survey (DHS) was performed in 85 developing nations including Bangladesh measuring the progress of population health and nutritional status. Since 1993, this survey has been conducted in Bangladesh in three-year intervals by the National Institute of Population Research and Training (NIPORT) of the Ministry of Health and Family Welfare [39]. Data files for this survey are open to access and use through the DHS program website [42]. To get access and use of the BDHS data, we registered on the MEASURE DHS (www.measuredhs.com) website and requested BDHS 2014 and 2017–18 datasets mentioning the study objectives. Afterward, DHS authorized us to download the required datasets within three working days. With the permission of the MEASURE DHS, data from the recent two waves of Bangladesh Demographic and Health Surveys (BDHS), 2014 and 2018 were downloaded, merged, and analyzed. We used children's data sets from each survey wave to extract the necessary variables included in this study [29, 39].

### Sampling design

The BDHS was implemented by a two-stage stratified cluster sampling for the household survey [29, 39]. The sampling framework of this survey was developed using the total list of enumeration areas (EAs) covering the entire country of the most recent census developed by the Bangladesh Bureau of Statistics (BBS). EAs are defined as geographic areas having an average of 120 households in each [43]. In the first stage, EAs were selected with probability proportional to the size of the EA. In the second stage, using a systemic sampling procedure a certain number of households on average (120) were selected from each EA with an equal probability to provide statistically valid estimates of health-related national-level indicators within the subgroups by divisions and place of residence. Ever-married women aged 15–49 years were interviewed for this survey to evaluate the utilization pattern of maternal healthcare services. The BDHS survey only included information on maternal health care services for women of the reproductive age group who had a live birth in three years preceding the survey. The rationale for excluding those with stillbirth outcomes in this study is the unavailability of maternal healthcare service information for stillbirth in the BDHS survey 2014 and 2017–18 [29, 39]. Additionally, information on attitudes towards domestic violence was collected for currently married women only in these surveys. The details of the sampling design of BDHS 2014 and BDHS 2018 are described elsewhere [29, 39].

## Study population

This study considered only currently married young mothers, aged 15–24 years who had a recent live birth in three years preceding the survey [44]. This allowed 2431 observations from BDHS 2014 and 2599 observations from BDHS 2018, after extracting and cleaning data available for the study. Among the studied young mothers, 32.4% and 48.3% of them in BDHS 2014 and 2018, respectively took four or more ANC visits. Moreover, 40.1% and 51.7% of young mothers delivered their recent birth at a health center in BDHS 2014 and BDHS 2018, respectively.

## Variables' characteristics

**Outcome variables.**   The study considered two binary outcome variables as indicators of maternal healthcare services utilization. The first outcome variable, utilization of four or more ANC visits categorized into two: 'yes' refers to those who had four or more ANC visits, and 'no' refers to those who took zero to three ANC visits. The second outcome variable is the place of delivery (PoD) which indicates whether the births were delivered at home or at a health center. If the birth took place at any health center, it is recorded as 'health-center-based delivery' and if the birth was delivered at home then it is recorded as 'home-based delivery'.

**Independent variables.**   The independent variable of 'young mothers' favorable attitude toward domestic violence, the key focus of the study, is considered a proxy variable for domestic violence [12, 39]. A composite variable was also measured through young mothers' attitudes towards justification of wife-beating or hitting by their husbands, which was grounded by their responses to five questions: 1) if she burns the food, 2) if she argues with her husband, 3) if she goes out without telling husband, 4) if she neglects the children, and 5) if she refuses to have sexual intercourse with husband. This variable was coded into two categories for analysis purposes: (1) 'favorable' refers to those who justified domestic violence with at least one of the above reasons, and (2) 'opposed' refers to otherwise.

A range of socio-economic and demographic variables was also included in the multivariate analysis. Region was coded into seven divisions for BDHS 2014: (1) Barisal, (2) Chittagong, (3) Dhaka, (4) Khulna, (5) Rajshahi, (6) Rangpur, and (7) Sylhet [39]. Mymensing was added as another administrative area along with these divisions only in the 2017–2018 BDHS [29]. Place of residence was coded into two categories: (1) rural and (2) urban. Households' wealth index was coded into three categories: (1) poor, (2) middle, and (3) rich. Educational status of young mothers and their husbands was coded into four categories: (1) no education, (2) primary who completed grade five, (3) secondary who completed grade 10, and (4) higher who attained above the secondary level education. Access to media was defined based on whether the respondents read newspapers/magazines, watched television, or listened to the radio. Thus, media exposure was classified into two: yes and no. Young mothers' decision-making capacity was coded into two: yes and no. If any respondent took part in making decisions solely or partially at least one of the issues related to own healthcare, key household purchases, and visiting her family members or relatives are categorized as "yes" in young mothers' autonomy variable and vice versa. The working status of mothers was categorized as (1) employed, and (2) unemployed. Birth order was classified into two: (1) first birth and (2) others.

## Statistical analyses

As the BDHS survey follows a two-stage stratified cluster sampling procedure, the two-level multilevel logistic model uses to examine the cluster effect among young mothers, and the random effect component is introduced for each cluster as the young mothers are nested into the clusters [45, 46]. The measures of association between two outcome variables and covariates

are assessed through chi-square tests. The independent variables that were significantly associated with the outcome variables are included in the regression model. For each of the two outcome variables, two models are considered- a null model without any explanatory variables and a complete model with all covariates.

Suppose $Y_{ij}$ be the binary outcome variable for $j^{th}$ individual in $i^{th}$ cluster where $j = 1,2,\ldots, n_i$ and $i = 1,2,\ldots,k$. Let $x_{ij} = (x_{ij1}, x_{ij2},\ldots,x_{ijp})'$ be the $p{\times}1$ vector of independent variables for $j^{th}$ individual in $i^{th}$ cluster and $\beta = (\beta_1, \beta_2,\ldots,\beta_p)'$ be the $p{\times}1$ vector of regression parameters. Then multilevel logistic regression can be written as

$$\ln\frac{\pi_{ij}}{1 - \pi_{ij}} = x'_{ij}\beta + u_i$$

Where $\pi_{ij} = E\,(Y_{ij}|u_i)$ and $u_i$ is random intercept where $u_i{\sim}N\,(0, \sigma_u^2)$. In this setup, the fixed effect of covariates can be expressed through an adjusted odds ratio, and random effect can be revealed through intra-cluster correlation (ICC) which means correlation among individuals within the same cluster. The denoted $\rho$ can be measured by the variance component of random intercept that is $\rho = \frac{\sigma_u^2}{\sigma_u^2 + \frac{\pi^2}{3}}$ [47].

## Results

The distribution of young mothers aged 15–24 years by background characteristics presented in Table 1 shows that the percentage of young mothers with four or more ANC visits was increased by about 16% (32.4% in 2014 and 48.3% in 2018) and health-center-based delivery was increased by about 12% (40.1% in 2014 and 51.7% in 2018) over the years. Young mothers'

**Table 1. Percentage distribution of young mothers aged 15–24 years in Bangladesh by selected background attributes, BDHS 2014 and 2018.**

| Characteristics | BDHS 2014 (n = 2431) | BDHS 2018 (n = 2599) |
|---|---|---|
| *At least 4 ANC* | | |
| Yes | 32.4 | 48.3 |
| No | 67.6 | 51.7 |
| *Place of Delivery (PoD)* | | |
| Health-center-based | 40.1 | 51.7 |
| Home-based | 59.9 | 48.3 |
| *Attitude towards domestic violence* | | |
| Favorable | 27.3 | 17.9 |
| Opposed | 72.7 | 82.1 |
| *Division* | | |
| Barisal | 11.8 | 11.1 |
| Chattogram | 19.5 | 16.7 |
| Dhaka | 16.6 | 14.4 |
| Khulna | 12.4 | 10.7 |
| Rajshahi | 12.9 | 10.9 |
| Rangpur | 13.0 | 11.0 |
| Sylhet | 13.9 | 13.1 |
| Mymensingh* | - | 12.2 |
| *Place of residence* | | |
| Rural | 68.3 | 67.7 |
| Urban | 31.7 | 32.3 |

(*Continued*)

**Table 1.** (Continued)

| Characteristics | BDHS 2014 (n = 2431) | BDHS 2018 (n = 2599) |
|---|---|---|
| *Wealth index* | | |
| Poor | 40.4 | 43.1 |
| Middle | 19.7 | 19.0 |
| Rich | 39.9 | 37.9 |
| *Young mothers' educational level* | | |
| No Education | 7.2 | 3.6 |
| Primary | 27.9 | 25.4 |
| Secondary | 54.5 | 52.1 |
| Higher | 10.3 | 18.9 |
| *Husband's educational level* | | |
| No Education | 35.2 | 30.3 |
| Primary | 45.0 | 46.1 |
| Secondary | 7.2 | 6.0 |
| Higher | 12.6 | 17.6 |
| *Access to media* | | |
| No | 36.3 | 35.8 |
| Yes | 63.7 | 64.2 |
| *Decision-making capacity* | | |
| No | 31.8 | 20.3 |
| Yes | 68.2 | 79.7 |
| *Working status* | | |
| No | 83.2 | 68.5 |
| Yes | 16.8 | 31.5 |
| *Birth order* | | |
| Otherwise | 35.2 | 37.6 |
| First birth | 64.8 | 62.4 |

* Mymensingh division has been included in BDHS 2018 wave only as a separate division.

opposing attitudes toward domestic violence were also increased by 9.4%, from 72.7% in 2014 to 82.1% in 2018. Whilst the proportion of poor young mothers was increased over the years, the proportion of rich young mothers was reduced. The percentages of young mothers and their husbands who had no education were decreased and a higher level of education for both of them was increased between 2014 and 2018. The percentage of young mothers with access to media was also increased over the years. Moreover, young mothers' decision-making capacity was increased by 11.5% and the proportion of employed young mothers was increased by about 15% between 2014 and 2018. The proportion of first birth orders was higher for young mothers in both years.

The overall utilization rates of four or more ANC visits and health-center-based delivery care among young mothers were increased between 2014 and 2018. However, the utilization rate was strikingly low for young mothers who had justified attitudes towards the reasons for domestic violence than those who had opposite attitudes, which indicates the negative effects of domestic violence on young mothers' utilization of four or more ANC and health-center-based delivery care (Table 2). It also shows that four or more ANC (29.5% in 2014 and 41.8% in 2018) and health-center-based delivery (30.4% in 2014 and 38.2% in 2018) service utilization rates among young mothers who had justified attitudes towards domestic violence if a

**Table 2. Percentage distribution of four or more ANC and health-center-based delivery care utilization among young mothers who have recent birth by reasons for domestic violence according to BDHS 2014 and 2018.**

| Domestic violence by husband is justified if women | % received at least 4 ANC | | % delivered birth at a health-center | |
|---|---|---|---|---|
| | BDHS 2014 | BDHS 2018 | BDHS 2014 | BDHS 2018 |
| *Goes out without telling husband* | 29.5 | 41.8 | 30.4 | 38.2 |
| *Neglects the children* | 29.1 | 41.0 | 30.7 | 42.2 |
| *Argues with husband* | 30.1 | 37.2 | 28.4 | 41.9 |
| *Refuses to have sex with husband* | 24.5 | 26.3 | 22.0 | 36.8 |
| *Burnt food* | 21.6 | 30.4 | 19.3 | 39.1 |

woman goes out without telling her husband were far behind the targets of Bangladesh's 4th Health, Population and Nutrition Sector Program (HPNSP) and Sustainable Development Goals (SDGs). Though the utilization rates of four or more ANC and health-center-based delivery care for young mothers who had justified attitudes towards the rest of the reasons for domestic violence were increased in 2018 than that were in 2014 but it tended to lag behind the aforementioned targets.

The percentage distribution of young mothers' who received four or more ANC and health-center-based delivery care by demographic and socioeconomic characteristics of the respondents is presented in Table 3. The table shows that young mothers' utilization of four or more ANC and health-center-based delivery care were varied substantially by their attitudes towards domestic violence in all categories, except for ANC use in 2014. Young mothers who had favorable attitudes towards domestic violence were significantly less likely to utilize four or more ANC compared to their counterparts who had opposed attitudes (p-value<0.001), particularly in 2018. Between 2014 and 2018, the utilization rate of four or more ANC was increased by 10.5% among young mothers who had a favorable attitude towards domestic violence and 16.8% who had an opposite attitude towards such violence. However, over the same period, the rate of health-center-based delivery care was increased more among young mothers who had favorable attitudes toward domestic violence (13.4%) than those who had opposite attitudes (9.8%).

There was a significant geographical variation in using four or more ANC and health-center-based delivery services (Table 3). In terms of using health-center-based delivery, the Sylhet division had been identified as a low-performing area of all divisions for both of the study points (24.3% and 42.8% respectively). Though Sylhet was a low-performing area considering the percentage of four or more ANC utilization in 2014 (23.1%), the situation became worsen for Mymensing in 2018 among all divisions (37.5%). Between 2014 and 2018, rural young women experienced a greater increase in using four or more ANC (16.6%) and health-facility-based delivery care (14.4%) than their counterparts' urban young women, 14.1% and 5.3% respectively. A subclass analysis by wealth status suggests that poor young women were less likely to use four or more ANC and health-facility-based delivery care than rich young women. The highest growth in receiving four or more ANC was noticed among young mothers from the middle wealth quintile by 19.0% (27.7% to 46.7%) followed by poor 17.0% (20.1% to 37.1%) and rich 14.7% (47.2% to 61.9%) between 2014 and 2018. However, the increase in the proportion of health-center-based delivery care was highest among poor young mothers followed by middle and rich young mothers.

Moreover, young mothers and their husbands with higher educational status were significantly more likely to use four or more ANC and health-center-based delivery care than those who had no education (Table 3). Whilst young mothers from all subclass of education were experienced greater use of four or more ANC and health-center-based delivery care between

**Table 3. Young mothers aged 15–24 years who received four or more ANC and health-center-based delivery by demographic and socioeconomic attributes, BDHS 2014 and 2018.**

| Variables | Received four or more ANC | | | | Delivered birth at a health center | | | |
|---|---|---|---|---|---|---|---|---|
| | BDHS 2014 | | BDHS 2018 | | BDHS 2014 | | BDHS 2018 | |
| | % | p-value | % | p-value | % | p-value | % | p-value |
| *Attitude toward domestic violence* | | | | | | | | |
| Favorable | 30.1 | 0.146 | 40.6 | <0.001 | 30.6 | <0.001 | 44.0 | <0.001 |
| Opposed | 33.2 | | 50.0 | | 43.6 | | 53.4 | |
| *Division* | | | | | | | | |
| Barisal | 26.6 | <0.001 | 37.8 | <0.001 | 27.6 | <0.001 | 44.8 | <0.001 |
| Chattogram | 28.1 | | 42.3 | | 37.8 | | 49.7 | |
| Dhaka | 37.2 | | 50.7 | | 47.6 | | 58.1 | |
| Khulna | 39.1 | | 62.1 | | 57.0 | | 64.3 | |
| Rajshahi | 30.7 | | 48.1 | | 46.6 | | 57.2 | |
| Rangpur | 43.0 | | 52.3 | | 39.2 | | 54.2 | |
| Sylhet | 23.1 | | 60.8 | | 24.3 | | 42.8 | |
| Mymensingh | - | | 37.5 | | - | | 44.3 | |
| *Place of residence* | | | | | | | | |
| Rural | 26.6 | <0.001 | 43.2 | <0.001 | 33.3 | <0.001 | 47.7 | <0.001 |
| Urban | 44.9 | | 59.0 | | 54.7 | | 60.0 | |
| *Wealth index* | | | | | | | | |
| Poor | 20.1 | <0.001 | 37.1 | <0.001 | 22.8 | <0.001 | 36.4 | <0.001 |
| Middle | 27.7 | | 46.7 | | 38.8 | | 50.9 | |
| Rich | 47.2 | | 61.9 | | 58.2 | | 69.4 | |
| *Young mothers' educational level* | | | | | | | | |
| No Education | 11.9 | <0.001 | 20.2 | <0.001 | 21.0 | <0.001 | 25.5 | <0.001 |
| Primary | 22.7 | | 34.3 | | 28.0 | | 33.7 | |
| Secondary | 35.7 | | 51.3 | | 43.5 | | 53.7 | |
| Higher | 55.4 | | 64.4 | | 67.7 | | 75.4 | |
| *Husband's educational level* | | | | | | | | |
| No Education | 21.8 | <0.001 | 35.5 | <0.001 | 25.8 | <0.001 | 36.3 | <0.001 |
| Primary | 33.8 | | 47.5 | | 41.2 | | 50.7 | |
| Secondary | 41.7 | | 55.4 | | 54.3 | | 63.7 | |
| Higher | 51.6 | | 70.2 | | 67.6 | | 76.6 | |
| *Access to media* | | | | | | | | |
| No | 20.2 | <0.001 | 34.3 | <0.001 | 23.4 | <0.001 | 36.1 | <0.001 |
| Yes | 39.3 | | 56.2 | | 49.6 | | 60.4 | |
| *Decision-making capacity* | | | | | | | | |
| No | 30.3 | 0.130 | 46.0 | 0.242 | 39.5 | 0.689 | 48.7 | 0.130 |
| Yes | 33.4 | | 48.9 | | 40.3 | | 52.4 | |
| *Working status* | | | | | | | | |
| No | 33.4 | 0.014 | 48.3 | 1.000 | 42.1 | <0.001 | 55.3 | <0.001 |
| Yes | 27.2 | | 48.3 | | 29.9 | | 43.8 | |
| *Birth order* | | | | | | | | |
| Otherwise | 26.9 | <0.001 | 41.6 | <0.001 | 28.6 | <0.001 | 39.8 | <0.001 |
| First birth | 35.4 | | 52.4 | | 46.3 | | 58.8 | |

2014 and 2018, the lowest increase occurred among young mothers with no education, 8.3%, and 4.5% respectively. Also, the utilization of four or more ANC and health-center-based delivery care were found substantially higher for young mothers who had access to media than

those who had no such access, in both years. Between 2014 and 2018, four or more ANC and health-center-based delivery care utilization were increased by 14.1% and 12.7% respectively for young mothers with access to media. Surprisingly, employed young mothers compared to unemployed were significantly less prevalent in using health-center-based delivery care in both years. Young mothers used four or more ANC and health-center-based delivery care more for their first birth compared to any other birth. However, there was a 14.7% increase in four or more ANC (26.9% to 41.6%) and an 11.2% increase in health-facility-based delivery care (28.6% to 39.8%) in terms of other births than the first birth between the two survey waves. However, women's decision-making capacity was not found significant in any of the years for both of the outcomes.

From the adjusted covariates (Table 4), the lower utilization of healthcare facilities for antenatal and delivery care was found to be associated with young mothers' favorable attitudes towards domestic violence in both of the survey years. More specifically, young mothers who had a favorable attitude towards domestic violence had 33.2% lower odds of getting health-center-based delivery care (OR = 0.668) in 2014 whereas the odds was lowered by 19.7% in 2018 (OR = 0.803). The utilization of four or more ANC and health-center-based delivery services significantly varied by division. Compared to other divisions, Rangpur experienced greater progress in the utilization of four or more ANC between 2014 (OR = 2.371) and 2018 (OR = 3.114). However, the Khulna division had the highest odds of receiving health-center-based delivery both in 2014 (OR = 3.706) and 2018 (OR = 1.753). In contrast, the lowest odds was found for Chittagong among all divisions in using four or ANC and health-center-based delivery care both in 2014 and 2018. Moreover, young mothers residing in urban areas were more likely to utilize four or more ANC in both periods than their counterparts in rural young women. However, the urban-rural difference was significantly reduced in using four or more ANC and health-center-based delivery care over the years.

The odds ratio of four or more ANC and health-center-based delivery care was increased with improved wealth status (Table 4). The same pattern in odds ratio was found across young mothers' and their husbands' educational attainment. These made the fact vivid that household wealth status and higher educational attainment of mothers and their husbands increased their utilization of four or more ANC and health-center-based delivery care. Moreover, access to media had a positive effect on using four or more ANC and health-center-based delivery services. Between 2014 and 2018, the likelihood of receiving four or more ANC and delivering birth at a health center was increased among young mothers with access to media, from (ORs) 1.261 to 1.638 and 1.409 to 1.566 respectively. Surprisingly, working young mothers were less likely to use health-center-based delivery care (OR = 0.631 and 0.731) in both years than their counterparts non-working young mothers. Moreover, there were 72.7% and 66.7% higher odds of having health-center-based delivery care in 2014 and 2018 respectively for the first birth of young mothers than the other births.

Table 4 also reported the cluster effect on four or more ANC and place of delivery service utilization based on the random effect for both of the survey years. Firstly, a separate null model was applied for each of the outcome variables to check whether the application of the multilevel logistic model in this study is justified. The result shows cluster variances for both outcome variables. The ICC obtained from the null model in 2018 data for four or more ANC was 0.24 which indicates that 24.0% of the total variation in receiving four or more ANC related to variation between clusters which was slightly low from 2014 BDHS (ICC = 0.25). On a contrary, the ICC estimated for the place of delivery from the null model was 0.20 which anticipated that 20.0% of the total variation in place of delivery was attributable to the clusters where mothers were residing which was higher than that in 2014 (ICC = 0.16). There was a substantial decrease in ICCs obtained separately for both dependent variables from the final

**Table 4. Fixed effect of several factors and random effects obtained from multilevel logistic regression model on at least 4 ANC and health-center-based delivery utilization.**

| Variables | At least 4 ANC | | | | Delivered birth at a health center | | | |
|---|---|---|---|---|---|---|---|---|
| | BDHS 2014 | | BDHS 2018 | | BDHS 2014 | | BDHS 2018 | |
| | AOR | p-value | AOR | p-value | AOR | p-value | AOR | p-value |
| *Attitude toward domestic violence* | | | | | | | | |
| Opposed | 1 | - | 1 | - | 1 | - | 1 | - |
| Favorable | 1.11 | 0.380 | 0.795 | 0.065 | 0.668 | 0.001 | 0.803 | 0.064 |
| *Division* | | | | | | | | |
| Barisal | 1 | - | 1 | - | 1 | - | 1 | - |
| Chittagong | 0.743 | 0.214 | 0.978 | 0.919 | 1.326 | 0.207 | 0.861 | 0.444 |
| Dhaka | 1.159 | 0.536 | 1.094 | 0.697 | 1.915 | 0.004 | 1.030 | 0.886 |
| Khulna | 1.415 | 0.162 | 2.310 | <0.001 | 3.706 | <0.001 | 1.753 | 0.008 |
| Rajshahi | 0.947 | 0.828 | 1.737 | 0.021 | 2.509 | <0.001 | 1.453 | 0.077 |
| Rangpur | 2.371 | <0.001 | 3.114 | <0.001 | 1.948 | 0.005 | 1.687 | 0.013 |
| Sylhet | 0.876 | 0.611 | 1.044 | 0.857 | 0.980 | 0.936 | 0.924 | 0.702 |
| Mymensingh | - | - | 1.695 | 0.024 | - | - | 1.033 | 0.876 |
| *Place of residence* | | | | | | | | |
| Rural | 1 | - | 1 | - | 1 | - | 1 | - |
| Urban | 1.741 | <0.001 | 1.674 | <0.001 | 1.728 | <0.001 | 1.106 | 0.382 |
| *Wealth index* | | | | | | | | |
| Poor | 1 | - | 1 | - | 1 | - | 1 | - |
| Middle | 1.215 | 0.217 | 1.128 | 0.375 | 1.430 | 0.015 | 1.212 | 0.136 |
| Rich | 2.240 | <0.001 | 1.621 | 0.001 | 2.140 | <0.001 | 2.198 | <0.001 |
| *Young mothers' educational level* | | | | | | | | |
| No Education | 1 | - | 1 | - | 1 | - | 1 | - |
| Primary | 1.917 | 0.019 | 2.079 | 0.017 | 1.048 | 0.842 | 1.247 | 0.421 |
| Secondary | 2.852 | <0.001 | 3.158 | <0.001 | 1.283 | 0.287 | 1.951 | 0.014 |
| Higher | 4.920 | <0.001 | 3.241 | <0.001 | 1.861 | 0.035 | 2.873 | <0.001 |
| *Husband's educational level* | | | | | | | | |
| No Education | 1 | - | 1 | - | 1 | - | 1 | - |
| Primary | 1.183 | 0.195 | 1.363 | 0.008 | 1.360 | 0.013 | 1.199 | 0.099 |
| Secondary | 1.194 | 0.416 | 1.496 | 0.062 | 1.601 | 0.027 | 1.460 | 0.072 |
| Higher | 1.398 | 0.094 | 2.702 | <0.001 | 2.419 | <0.001 | 2.117 | <0.001 |
| *Access to media* | | | | | | | | |
| No | 1 | - | 1 | - | 1 | - | 1 | - |
| Yes | 1.261 | 0.089 | 1.638 | <0.001 | 1.409 | 0.008 | 1.566 | <0.001 |
| *Working status* | | | | | | | | |
| No | 1 | - | - | - | 1 | - | 1 | - |
| Yes | 0.846 | 0.316 | - | - | 0.631 | <0.001 | 0.731 | 0.003 |
| *Birth order* | | | | | | | | |
| Otherwise | 1 | - | 1 | - | 1 | - | 1 | - |
| First birth | 1.120 | 0.252 | 1.181 | 0.093 | 1.727 | 0.001 | 1.667 | <0.001 |
| **Intra-cluster Correlation (ICC)** | | | | | | | | |
| Null model | 0.25 | | 0.24 | | 0.16 | | 0.20 | |
| Full model | 0.16 | | 0.15 | | 0.27 | | 0.07 | |

model. The cluster variation observed in the null model can mostly be explained by consideration of the effect of independent variables on four more ANC and health-center-based delivery services.

## Discussion

This paper examines the association between attitude towards domestic violence and the utilization of four or more ANC and health-center-based delivery care along with the presence of cluster variation in Bangladesh. This pattern should be a major concern to achieving universal maternal healthcare coverage, as inequity and domestic violence towards women are yet important barriers indeed in using four or more ANC and health-center-based delivery care. To our knowledge, this study for the first time in Bangladesh uses the recent BDHS datasets (2014 and 2018) in analyzing the effect of the favorable attitude of young women towards domestic violence on four or more ANC and health-center-based delivery care services utilization. The overall utilization of four or more ANC and health-center-based delivery care among young mothers has increased by 10.5% and 13.4% respectively in 2018 than that was in 2014. However, the utilization of four or more ANC and health-center-based delivery care is found substantially low for young mothers who had favorable attitudes towards several reasons for domestic violence. Moreover, the utilization of four or more ANC and health-center-based delivery care among young mothers is far behind the targets of 4th HPNSP and SDGs.

This study demonstrates that young mothers who had favorable attitudes towards domestic violence were less likely to utilize four or more ANC and health-center-based delivery care than those who had the opposite attitudes. The limited use of four or more ANC in 2018 and health-center-based delivery care in 2014 and 2018 was strongly associated with young mothers' favorable attitude towards domestic violence. These findings are consistent with previous studies conducted in the context of Asian and African countries [21, 22, 48–50]. In Bangladesh, women's favorable attitude towards domestic violence is associated with their inferior social position, inferiority, negligence, poor educational status, lower sense of entitlement, lack of self-esteem, and lack of awareness [24, 35, 38, 51–53]. Therefore, domestic violence is an important social as well as public health concern in improving the utilization of quality ANC and health-center-based delivery care in Bangladesh and some other countries like India, Tanzania, Ethiopia, etc.

Moreover, this study finds the presence of cluster variation in the utilization of four or more ANC and health-center-based delivery care for both of the survey years. Young mothers from disadvantaged communities have inadequate utilization of four or more ANC and health-center-based delivery care. This finding is largely found consistent with a study conducted in Ethiopia which reported that mothers from communities with high amenities have greater access to and utilization of the WHO-recommended number of ANC visits and hospital-based delivery care than those who live in communities with low amenities. The variations in using four or more ANC and health-center-based delivery care among clusters due to geographic, cultural, and socio-economic differences are important public health concerns in Bangladesh that need to be addressed to make healthcare services more accessible to all women.

The present study suggests that along with domestic violence, multiple socio-economic and demographic covariates are significantly associated with the lower utilization of four or more ANC and health-center-based delivery care. Geographical variation is also found significant for reducing the use of four or more ANC and health-center-based delivery care. Rural young mothers are less likely to utilize four or more ANC and health-center-based delivery care compared to urban young mothers which is consistent with earlier studies [31, 33, 35, 54]. The utilization of four or more ANC and health-center-based delivery care also significantly vary by division. The utilization rate of four or more ANC among young mothers of the Sylhet division has increased remarkably in 2018 which was previously identified as low performing division in terms of using ANC visits [29, 39]. However, among the divisions, young mothers in the

Sylhet division are less likely to utilize health-center-based delivery care which is found consistent with previous studies [55, 56]. Possible causes of such limited utilization of health services are poor transportation, geographical remoteness, and inadequate counseling about the importance of health-center-based delivery care [21, 54, 56].

Our study in line with other studies identified that poor young mothers are less likely to use four or more ANC and health-center-based delivery care compared to rich young mothers in both survey periods [35, 40, 57]. This finding highlights that despite the socio-economic progress, Bangladesh is still experiencing inequities in the utilization of healthcare services [35]. We also found that the utilization of four or more ANC and health-center-based delivery care are associated with young mothers' education and their husband's education which are identical to earlier studies conducted in Bangladesh [31, 35]. Both young mothers and their husbands with a higher level of educational status appear with more awareness and knowledgeable regarding the quality of maternal healthcare which increases young mothers' likelihood of using that services for antenatal and delivery care [35, 48, 58, 59]. Mass media exposure also contributes to increasing young mothers' utilization of four or more ANC and health-center-based delivery care which is also reported in previous studies [31, 36, 58]. Mass media is a great source of diffusing information that increases awareness of individuals through behavior change communication and develops a tendency among them to adopt a positive or new behavior.

Surprisingly, employed young mothers in our study are found less likely to utilize four or more ANC and health-center-based delivery in both of the survey years. Despite being employed, Bangladeshi young women have limited liberty to exercise their own decision and ability to spend money on their healthcare. Furthermore, employed young mothers sometimes feel uncomfortable using these services due to shyness and family restrictions [38]. Also, under-reporting of health center-based delivery, especially for private hospitals or clinics might be contributing to such lower utilization of services among young mothers [16, 23, 33, 51, 60]. This study also reveals young mothers with first pregnancies are more likely to utilize those services compared to their other births. First pregnancy may be viewed with delicacy due to the new experience and excitement of the couples which increases their utilization of ANC and health-center-based delivery care, especially for young mothers [33, 54, 61].

## Conclusions

The findings of the study demonstrate that young mothers with favorable attitudes towards domestic violence are less likely to use four or more ANC and health-center-based delivery care than mothers with opposite attitudes. A strong variation in the clusters, which is identified in our study, is also associated with lower utilization of four or more ANC and health-center-based delivery care. In view of the lower utilization of such services among mothers who justified domestic violence, the health system of Bangladesh may initiate social campaigning and counseling intervention for young women and their intimate partners highlighting the benefits of using quality maternal care and the health consequences of domestic violence. Considering the variation among clusters, the coverage of the maternity allowance program should be extended; for universal maternal healthcare coverage in Bangladesh [28].

The major strength of our study is the consideration of nationally representative recent waves of BDHS data (2014 and 2018) to identify the association between attitudes towards domestic violence and utilization of four or more ANC and health-center-based delivery care among young Bangladeshi women. Moreover, the cluster variations in using four or more ANC and health-center-based delivery care have been assessed rigorously in this study. Our findings suggest that there is a scope for further study to identify the cluster variations

considering the community-level features and extend the analysis to a three-level multilevel analysis. The study may have recall bias though the effect of recall bias is minimized by considering the young mothers only with last birth in the preceding three years of the surveys. This study using regression analysis identified the associations between the predictors and outcome measures only, not the causal relationship.

## Acknowledgments

We are indebted to ICF International, Rockville, Maryland, USA, for giving us access to the 2014 and 2017–2018 Bangladesh Demographic and Health Survey datasets. We would like to thank the editor and the anonymous reviewers for their constructive insights and guidelines.

## Author Contributions

**Conceptualization:** Sawkia Afroz, Tasmiah Sad Sutopa, Md Rabiul Haque.

**Data curation:** Tasmiah Sad Sutopa.

**Formal analysis:** Sawkia Afroz, Tasmiah Sad Sutopa.

**Methodology:** Sawkia Afroz, Tasmiah Sad Sutopa, Md Rabiul Haque.

**Supervision:** Md Rabiul Haque.

**Writing – original draft:** Sawkia Afroz, Tasmiah Sad Sutopa, Md Rabiul Haque.

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
