## [Decision Letter · Decision Letter 0]

21 Oct 2021

PONE-D-21-14055Young mothers’ attitudes towards domestic violence and their maternal healthcare services utilization in Bangladesh: A multilevel cluster analysisPLOS ONE

Dear Dr. Haque,

Thank you for submitting your manuscript to PLOS ONE. After careful consideration, we feel that it has merit but does not fully meet PLOS ONE’s publication criteria as it currently stands. Therefore, we invite you to submit a revised version of the manuscript that addresses the points raised during the review process.

There are some minor issues identified by the reviewers those need to be fixed before taking final decision.

We look forward to receiving your revised manuscript.

Kind regards,

Enamul Kabir

Academic Editor

PLOS ONE

Journal Requirements:

"No - The funders had no role in study design, data collection and analysis, decision to publish, or preparation of the manuscript."

Reviewers' comments:

Reviewer's Responses to Questions

**Comments to the Author**

1. Is the manuscript technically sound, and do the data support the conclusions?

Reviewer #1: Yes

Reviewer #2: Yes

2. Has the statistical analysis been performed appropriately and rigorously? 

Reviewer #1: Yes

Reviewer #2: Yes

3. Have the authors made all data underlying the findings in their manuscript fully available?

Reviewer #1: Yes

Reviewer #2: Yes

4. Is the manuscript presented in an intelligible fashion and written in standard English?

Reviewer #1: Yes

Reviewer #2: Yes

5. Review Comments to the Author

Reviewer #1: It is a well written paper; I suggest just a few comments for more improvement. The introduction section is well written and nicely structured. Few comments, the authors could review for repetitive things such as still birth and death of fetus as an indirect consequence of violence against women during pregnancy.

Methods section:

1. Data source is described to be the DHS. A little more specific details of how the data were obtained, what procedures were followed and long with what, if, a letter was required and for which datasets/variables data was accessed and how long it took would give readers more picture of the DHS data access.

2. The authors mentioned including only currently married young mothers, aged 15-24 years who had a recent live birth in three years preceding the survey. A detailed explanation why would be important: for example what was the rationale of excluding those with stillbirth outcome?.

Results:

Informative a bit lengthy. Also could use language and editorial revisions. some of the sentences were too long and difficult to understand the intention/message.

Conclusion:

Appears too long. Suggest summarizing.

Reviewer #2: The study is a good piece of exploration in the BDHS data on an important topic. The authors explored utilization of antenatal care and health-center-based delivery by young mothers in Bangladesh along with an evaluation of its association with submissive attitude of young mothers towards domestic violence. The overall methodology, analysis looks sound. However, there is necessity and room for improvement in interpretation.

I've highlighted the areas where I have put queries or suggestions in the manuscript draft.

There are minor grammatical mistakes (incorrect use of passive voice in several areas) in the draft which should be corrected before re-submission. Note that I didn't highlight those areas.

6. PLOS authors have the option to publish the peer review history of their article (what does this mean?). If published, this will include your full peer review and any attached files.

Reviewer #1: No

Reviewer #2: No

---

## [Author Response · Author response to Decision Letter 0]

1 Feb 2022

1 February 2022

Dr. Enamul Kabir

Academic Editor

PLOS ONE

Cambridge, United Kingdom

Subject: Submission of revised manuscript PONE-D-21-14055 for publication into PLOS ONE

Dear Dr. Kabir

We Sawkia Afroz, Tasmiah Sad Sutopa, and Md Rabiul Haque, are grateful for providing constructive feedback to our submission PONE-D-21-14055 and giving us the opportunity for resubmission. We have reviewed the full manuscript and addressed the comment very carefully. 

Reviewer #1: It is a well written paper; I suggest just a few comments for more improvement. The introduction section is well written and nicely structured. Few comments, the authors could review for repetitive things such as still birth and death of fetus as an indirect consequence of violence against women during pregnancy.

Response: The text below has been incorporated in page 3, line numbers 42-47. 

“Evidence also suggested that in developing countries like Bangladesh miscarriage, preterm birth, stillbirth, low birth weight in newborns and maternal deaths followed by maternal morbidity are indirect consequences of domestic violence against women during pregnancy.” 

Methods section:

1. Data source is described to be the DHS. A little more specific details of how the data were obtained, what procedures were followed and long with what, if, a letter was required and for which datasets/variables data was accessed and how long it took would give readers more picture of the DHS data access.

Response: The following text has been incorporated in page 5, line numbers 102-110. 

“In order to get access and use of the BDHS data, we registered in the MEASURE DHS [Monitoring and Evaluation to Assess and Use Results, Demographic and Health Surveys] (www.measuredhs.com) website and requested for BDHS 2014 and 2017-18 datasets mentioning the study objectives. Afterward, DHS authorized us to download the required datasets within three working days. With the permission of the MEASURE DHS, data from the recent two waves of Bangladesh Demographic and Health Surveys (BDHS), 2014 and 2018 were downloaded, merged, and analyzed. We used children’s data sets from each survey wave to extract necessary variables included in this study (29,39).” 

2. The authors mentioned including only currently married young mothers, aged 15-24 years who had a recent live birth in three years preceding the survey. A detailed explanation why would be important: for example what was the rationale of excluding those with stillbirth outcome?

Response: In response to this point, the following text has been incorporated in page 6, line numbers 125-132.

“The BDHS survey only included the information of maternal health care services for women of the reproductive age group who had a live birth in 3 years preceding the survey. The rationale for excluding those with stillbirth outcomes is the unavailability of maternal healthcare service information for stillbirth in the BDHS survey 2014 and 2017-18. Additionally, information on attitude towards domestic violence was collected for currently married women only in these surveys. However, this study in line with the objectives considered only currently married young mothers, aged 15-24 years who had a recent live birth in three years preceding the survey.” 

Results: 

Informative a bit lengthy. Also could use language and editorial revisions. Some of the sentences were too long and difficult to understand the intention/message.

Response: Thanks for spotting the errors. We have gone through the manuscript rigorously and revised the long sentences to make them clear and understandable.

Conclusion:

Appears too long. Suggest summarizing.

Response: In response, we have reviewed and summarized the conclusion part.

Reviewer #2: The study is a good piece of exploration in the BDHS data on an important topic. The authors explored utilization of antenatal care and health-center-based delivery by young mothers in Bangladesh along with an evaluation of its association with submissive attitude of young mothers towards domestic violence. The overall methodology, analysis looks sound. However, there is necessity and room for improvement in interpretation.

I've highlighted the areas where I have put queries or suggestions in the manuscript draft.

There are minor grammatical mistakes (incorrect use of passive voice in several areas) in the draft which should be corrected before re-submission. Note that I didn't highlight those areas.

Responses: Based on reviewer’s recommendations, the highlighted texts have been modified as below: 

1. The highlighted words “strongly correlated” has been replaced with “associated” in page 14 line number 289 based on reviewer’s reommendation.

2. The highlighted text “strong association between domestic violence and the utilization” has been modified as “---association between attitude towards domestic violence and the utilization---” in page 16 line number 330.

3. The highlighted text “analyzing the effects of domestic violence on young mothers’ four or more ANC and health-center-based delivery care” has been modified as “---analyzing the effect of favorable attitude of young mothers towards domestic violence on four or more ANC and health-center-based delivery care---” in page no 16 line numbers 337-338.

4. The highlighted text “the utilization of four or more ANC and health-center-based delivery

 services were increased less over the years among young mothers who had favorable attitudes towards domestic violence than those who had the opposite attitudes” has been modified as “---young mothers who had favorable attitudes towards domestic violence were less likely to utilize four or more ANC and health-center-based delivery care than those who had the opposite attitudes---” in page no 16 line numbers 345-348.

5. In response to the reviewrs’ recommendation, the definition of “domestic violence” has been elaborated in the methods section (page no 7 line number 155-163) as below:

“The key focus of this study is on the independent variable ‘young mothers’ favorable attitude toward domestic violence which is considered as a proxy variable for domestic violence. A composite variable was measured through young mothers’ attitudes towards justification of wife-beating or hitting by their husbands, which was grounded by their responses to five questions: 1) if she burns the food, 2) if she argues with husband, 3) if she goes out without telling husband, 4) if she neglects the children, and 5) if she refuses to have sexual intercourse with husband. However, this variable for analysis purposes was coded into two categories: (1) ‘favorable’ refers to those who justified domestic violence with at least one of the above reasons, and (2) ‘opposed’ refers to otherwise.”

6. The highlighted text “Besides due to shyness, scarcity of female healthcare practitioners, and restriction imposed by husbands and family members many employed young mothers sometimes feel uncomfortable using the WHO-required number of ANC and hospital-based delivery care has been modified as “Furthermore, employed young mothers sometimes feel uncomfortable in using these services due to shyness and family restrictions (38). Also, under-reporting of health center-based delivery, especially for private hospitals or clinics might be contribute to such lower utilization of services among young mothers (16, 23, 33, 51, 60)” in page no 18 line numbers 399-405.

7. Considering the reviewers' recommendation, we have addressed the grammatical errors throughout the manuscript and incorrect use of passive voice in some areas.

The manuscript has not been published and is not under consideration elsewhere. We have no conflicts of interests to disclose and no reservation for any reviewers. 

All authors have seen and approved the revised version of the manuscript. 

Sincerely, 

Corresponding Author (PONE-D-21-14055)

Md Rabiul Haque, PhD 

Professor, Department of Population Sciences 

University of Dhaka, Bangladesh

---

## [Decision Letter · Decision Letter 1]

11 Mar 2022

PONE-D-21-14055R1Young mothers’ attitudes towards domestic violence and their maternal healthcare services utilization in Bangladesh: A multilevel cluster analysisPLOS ONE

Dear Dr. Haque,

Thank you for submitting your manuscript to PLOS ONE. After careful consideration, we feel that it has merit but does not fully meet PLOS ONE’s publication criteria as it currently stands. Therefore, we invite you to submit a revised version of the manuscript that addresses the points raised during the review process.

We look forward to receiving your revised manuscript.

Kind regards,

Enamul Kabir

Academic Editor

PLOS ONE

Journal Requirements:

Reviewers' comments:

Reviewer's Responses to Questions

**Comments to the Author**

1. If the authors have adequately addressed your comments raised in a previous round of review and you feel that this manuscript is now acceptable for publication, you may indicate that here to bypass the “Comments to the Author” section, enter your conflict of interest statement in the “Confidential to Editor” section, and submit your "Accept" recommendation.

Reviewer #2: All comments have been addressed

2. Is the manuscript technically sound, and do the data support the conclusions?

Reviewer #2: Yes

3. Has the statistical analysis been performed appropriately and rigorously? 

Reviewer #2: Yes

4. Have the authors made all data underlying the findings in their manuscript fully available?

Reviewer #2: Yes

5. Is the manuscript presented in an intelligible fashion and written in standard English?

Reviewer #2: No

6. Review Comments to the Author

Reviewer #2: First, I want to thank the authors for the changes they made in the revised version. However, I still think the English of the manuscript is poor and needs revision. The authors are suggested to increase clarity of their writing by revising it with a native speaker or an expert user of English or a professional grammar editing software/service.

7. PLOS authors have the option to publish the peer review history of their article (what does this mean?). If published, this will include your full peer review and any attached files.

Reviewer #2: No

---

## [Author Response · Author response to Decision Letter 1]

9 Apr 2022

We have uploaded the required files (docs)

---

## [Editor Report · Decision Letter 2]

22 Apr 2022

Young mothers’ attitudes towards domestic violence and their maternal healthcare services utilization in Bangladesh: A multilevel cluster analysis

PONE-D-21-14055R2

Dear Dr. Haque,

We’re pleased to inform you that your manuscript has been judged scientifically suitable for publication and will be formally accepted for publication once it meets all outstanding technical requirements.

Kind regards,

Enamul Kabir

Academic Editor

PLOS ONE
---

## [Editor Report · Acceptance letter]

21 Jul 2022

PONE-D-21-14055R2 

Young mothers’ attitudes towards domestic violence and their maternal healthcare services utilization in Bangladesh: A multilevel cluster analysis 

Dear Dr. Haque:

I'm pleased to inform you that your manuscript has been deemed suitable for publication in PLOS ONE. Congratulations! Your manuscript is now with our production department. 

Kind regards, 

on behalf of

Dr. Enamul Kabir 

Academic Editor

PLOS ONE